# Social interaction-induced activation of RNA splicing in the amygdala of microbiome-deficient mice

Roman M Stilling[1,2†]*, Gerard M Moloney[1,2], Feargal J Ryan[1,3], Alan E Hoban[1,2], Thomaz FS Bastiaanssen[1,2], Fergus Shanahan[1], Gerard Clarke[1,4], Marcus J Claesson[1,3], Timothy G Dinan[1,4], John F Cryan[1,2]*

[1]APC Microbiome Institute, University College Cork, Cork, Ireland; [2]Department of Anatomy and Neuroscience, University College Cork, Cork, Ireland; [3]School of Microbiology, University College Cork, Cork, Ireland; [4]Department of Psychiatry and Neurobehavioural Science, University College Cork, Cork, Ireland

**\*For correspondence:**
roman.stilling@gmail.com (RMS);
j.cryan@ucc.ie (JFC)

**Present address:** †German Primate Center, Goettingen, Germany

**Abstract** Social behaviour is regulated by activity of host-associated microbiota across multiple species. However, the molecular mechanisms mediating this relationship remain elusive. We therefore determined the dynamic, stimulus-dependent transcriptional regulation of germ-free (GF) and GF mice colonised post weaning (exGF) in the amygdala, a brain region critically involved in regulating social interaction. In GF mice the dynamic response seen in controls was attenuated and replaced by a marked increase in expression of splicing factors and alternative exon usage in GF mice upon stimulation, which was even more pronounced in exGF mice. In conclusion, we demonstrate a molecular basis for how the host microbiome is crucial for a normal behavioural response during social interaction. Our data further suggest that social behaviour is correlated with the gene-expression response in the amygdala, established during neurodevelopment as a result of host-microbe interactions. Our findings may help toward understanding neurodevelopmental events leading to social behaviour dysregulation, such as those found in autism spectrum disorders (ASDs).

DOI: https://doi.org/10.7554/eLife.33070.001

## Introduction

The tight association that animals have with the trillions of microbes that colonise them is the result of a long evolutionary history. Although we only very recently started to understand the intimate relationship between microbes and host physiology, including brain function, it is now well accepted that host neurodevelopment, brain function and behaviour are regulated by presence and activity of the host-associated microbiota (*Collins et al., 2012*; *Cryan and Dinan, 2012*; *Foster et al., 2017*; *Lyte, 2013*; *Mayer et al., 2014a*; *Sampson and Mazmanian, 2015*; *Sharon et al., 2016*).

One of the most recurrent and evolutionary conserved behaviours observed to be influenced by the microbiota, both during a host's lifetime as well as on evolutionary time scales, is host social behaviour (*Arentsen et al., 2015*; *Crumeyrolle-Arias et al., 2014*; *de Theije et al., 2014*; *Desbonnet et al., 2014*; *Ezenwa et al., 2012*; *Gacias et al., 2016*; *Kwong et al., 2017*; *Leclaire et al., 2017*; *Lewin-Epstein et al., 2017*; *Montiel-Castro et al., 2013*; *Montiel-Castro et al., 2014*; *Parashar and Udayabanu, 2016*; *Sharon et al., 2010*; *Snyder-Mackler et al., 2016*; *Stilling et al., 2014a*; *Theis et al., 2013*; *Tung et al., 2015*). Importantly, a growing body of data in healthy volunteers and patient populations is emerging, indicating microbial influences also translate to human emotional behaviours (*Tillisch et al., 2013*) and have been suggested to play a role in neurodevelopmental disorders such as autism spectrum disorders (ASDs) (*Stilling et al.,*

**eLife digest** In our bodies, there are at least as many microbial cells as human cells. These microbes, known collectively as the microbiome, influence the activity of our brain and also our behaviour. Studies in species from insects to primates have shown that the microbiome affects social behaviour in particular. For example, germ-free mice, which grow up in a sterile environment and thus have no bacteria in or on their bodies, are less sociable than normal mice.

For animals to show behaviours such as social interaction, cells in specific regions of the brain must change the activity of their genes. These brain regions include the amygdala, which is part of the brain's emotion processing network, and also contributes to fear and anxiety responses. Stilling et al. set out to determine whether gene activity in the amygdala during social interaction differs between germ-free mice and those with a normal microbiome.

Stilling et al. placed each mouse into a box with three chambers. One chamber contained an unfamiliar mouse while another contained an inanimate object. Germ-free mice were less sociable and spent less time than control animals interacting with the unfamiliar mouse. Before entering either test chamber, the germ-free animals showed signs of excessive activity in the amygdala. During social interaction, they displayed a strikingly different pattern of gene activity in this brain region compared to controls. In particular, they had increased levels of a process called alternative splicing. This process enables cells to produce many different proteins from a single gene.

These results reveal one of the steps leading from absence of bacteria during brain development to reduced sociability in adulthood in mice. Increases in gene activity in the amygdala may provide clues to the processes underlying reduced sociability in people with autism spectrum disorders. This new study thus deepens our understanding of the link between the microbiome and brain health.
DOI: https://doi.org/10.7554/eLife.33070.002

*2014a*; *Hsiao et al., 2013*; *Kang et al., 2017*; *Mayer et al., 2014b*; *Strati et al., 2017*) and schizo-phrenia (*Dinan et al., 2014*). Just very recently, it was shown in two independent mouse models of autism that the microbiota is necessary for expression of autistic-like symptoms in these models (*Golubeva et al., 2017*; *Kim et al., 2017*). It is, however, still largely unclear how and where the microbiota influence brain function and which mechanisms mediate changes in behaviour.

While the amygdala is critically involved in anxiety and fear-related behaviours and memory, it is also a well-established key emotional brain centre for evaluating and responding to social stimuli in humans and other mammals (*Allsop et al., 2014*; *Amaral, 2003*; *Kliemann et al., 2012*; *Noonan et al., 2014*; *Phelps and LeDoux, 2005*; *Sabatini et al., 2007*; *Sallet et al., 2011*). As such, neuropsychiatric disorders characterised by social deficits (including autism spectrum and anxiety disorders) are associated with structural and functional changes in the amygdala (*Amaral and Corbett, 2003*; *Baron-Cohen et al., 2000*; *Monk et al., 2010*; *Schultz, 2005*). Evidence for a role of the microbiota in regulating amygdala function is emerging (*Tillisch et al., 2013*; *Hoban et al., 2018*; *Stilling et al., 2015*; *Luczynski et al., 2016a*; *Hoban et al., 2017*) but this has been largely descriptive.

Germ-free (GF) mice, lacking microbial colonisation throughout development, are a well-estab-lished and essential tool spearheading the characterisation of microbiota-host interactions in regulat-ing development and physiological and behavioural parameters in the host (*Desbonnet et al., 2014*; *Hsiao et al., 2013*; *Bäckhed et al., 2007*; *Bercik et al., 2011*; *Clarke et al., 2013*; *Diaz Heijtz et al., 2011*; *McVey Neufeld et al., 2013*; *Neufeld et al., 2011*; *Ridaura et al., 2013*; *Luczynski et al., 2016*). By colonising GF mice at weaning age (exGF), developmental effects can be distinguished from dynamic, reversible effects of a functional microbiota. Previous studies, focussed on the ability of post-weaning colonisation of formerly GF rodents to reverse behavioural deficits, have produced mixed results with some phenotypes being reversible while others seemed to be developmentally programmed (see (*Stilling et al., 2014b*) for review). Thus, the mechanistic under-pinnings of behavioural changes in GF mice and how they are controlled by early colonisation during development are still elusive.

We here provide evidence that the microbiota is a critical regulator of social interaction-induced gene expression in the amygdala. Using an unbiased, genome-wide approach to determine gene

expression in the amygdala by paired-end, stranded, ribodepleted RNA-sequencing together with a comprehensive downstream analysis pipeline, we studied alterations in the amygdala transcriptome in response to a social interaction stimulus. We find a unique transcriptional response in GF mice that involves upregulation of the splicing machinery, which is able to partially compensate for impairments in neuronal plasticity signalling during social interaction in these animals.

## Results

### Germ-free mice display impaired sociability behaviour with high inter-individual variability

Mice lacking any interaction with microorganisms throughout development have a range of behavioural phenotypes (*Luczynski et al., 2016*), including memory impairments and altered anxiety behaviour (*Hoban et al., 2018*; *Gareau et al., 2011*).

Here, we subjected conventional mice (CON-SI), germ-free mice (GF-SI), and germ-free mice colonized after weaning (exGF-SI) to a social stimulus (the three-chamber social interaction test (3CSIT), based on *Nadler et al., 2004*) and measured the time during which the test mice interacted with a conventional conspecific male mouse or a non-social object (*Figure 1A*, experimental design and workflow). As previously reported (*Desbonnet et al., 2014*; *Buffington et al., 2016*), on average the group of GF-SI mice showed significantly decreased interaction with a conspecific compared with controls and colonized animals, while interaction with the non-social object was similar among the three groups (*Figure 1B–C*). Notably, we found high inter-individual variability in the GF-SI group for the time interacting with the conspecific, ranging from approximately control levels (>300 s) to as little as 60 s (*Figure 1B*). However, the distribution passed the D'Agostino and Pearson omnibus normality test, the Shapiro-Wilk normality test and the KS normality test, all at alpha = 0.05. Time spent with the non-social object was similarly variable in all three groups (*Figure 1C*).

### Social interaction induces highly distinct gene expression patterns in the amygdala of germ-free mice

Since we have previously shown that social interactions in the mouse 3CSIT behavioural paradigm reliably activates stimulus-dependent genes expression in the amygdala (*Stilling et al., 2015*), we hypothesised that we would observe changes in gene expression in this brain region as a function of colonisation status and, especially, social experience. Thus we sought to identify potential molecular pathways involved in mediating microbiome-to-brain signalling by systematically comparing gene expression patterns in the amygdala of mice that were exposed to social environmental stimulation (social interaction, SI) mice. To this end, we retrieved RNA from naïve animals and 1 hr after the social experience for CON, GF and exGF animals and performed a highly comprehensive type of RNA-sequencing, using paired-end RNA libraries that retains information on which DNA strand was transcribed and includes also non-polyadenylated RNA species. This way our analysis incorporated gene expression changes at baseline and in response to stimulus-induced transcription of both, mRNAs and long non-coding RNAs (lncRNAs) that lack a poly-A tail.

In a first step, we performed analysis of differential gene expression on all meaningful pairwise comparisons (*Table 1*). When comparing the number of differentially expressed genes (DEGs, both up- and downregulated) between naïve animals of different colonisation status, we find the highest number of DEGs in the CON-GF comparison (*Figure 2A*, *Supplementary file 1*), as we have previously reported in an independent study (*Stilling et al., 2015*). Interestingly, when comparing naïve vs. stimulated animals the response in terms of number of DEGs was strongest in GF animals by far, providing evidence for the amygdalar transcriptomes to diverge between the three groups upon environmental stimulation by social interaction. Transcriptional divergence driven by neuronal activity is further supported by the number of DEGs when comparing the SI groups among each other, as these numbers are substantially (4–9 times) higher than DEG numbers for naïve comparisons (*Figure 2A*). Increasing stringency of the analysis (log2(fold-change) > |±0.5|) had little relative effect on these results (*Figure 2—figure supplement 1A*).

Next we looked for overlapping DEGs between pairwise comparisons. Because meaningful Venn diagrams are limited to four or five comparisons, to identify groups of genes that are regulated in multiple comparisons, we plotted the presence or absence for each of the 4522 non-redundant

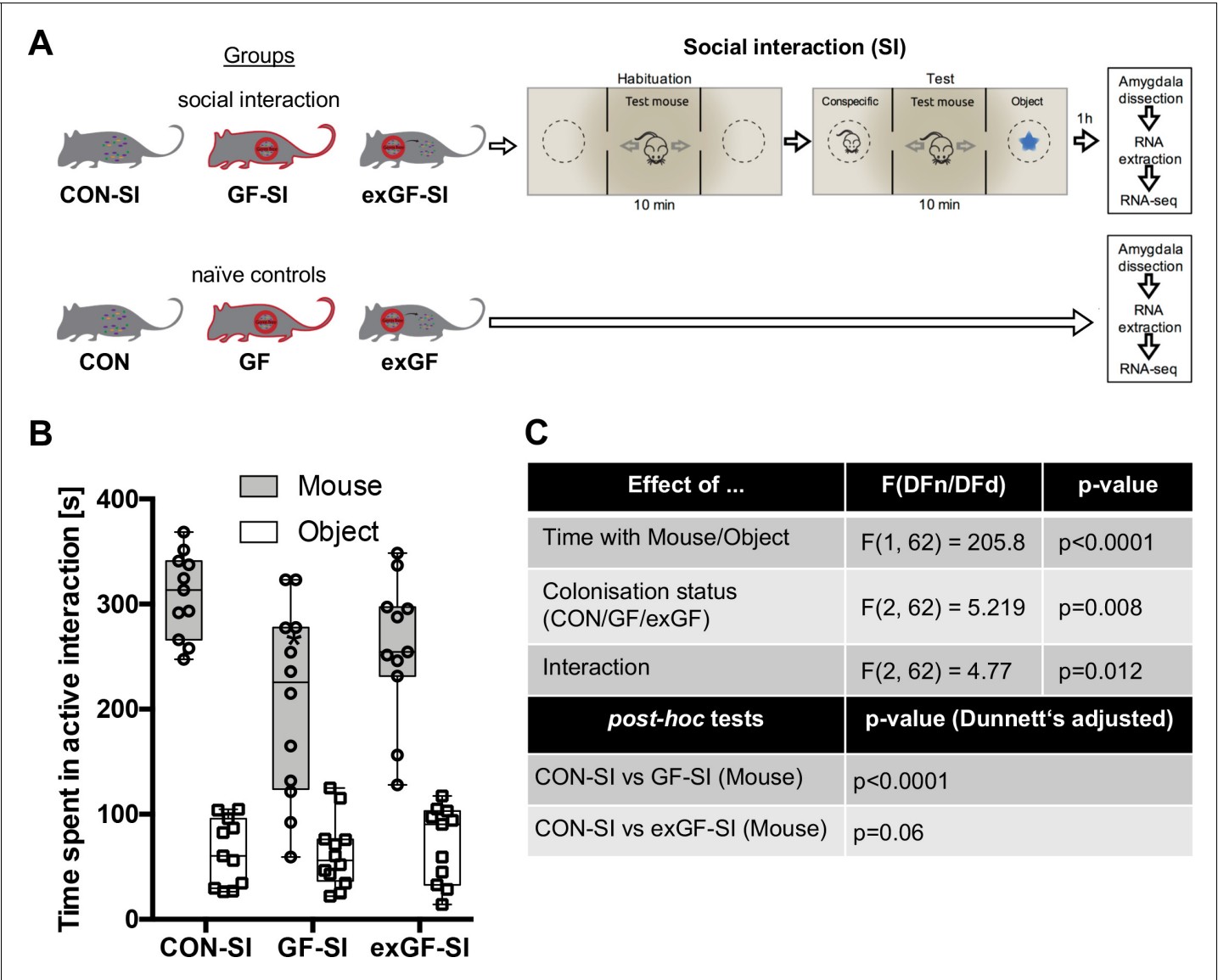

**Figure 1.** Germ-free (GF) mice display reduced social interaction. (**A**) Experimental design. (**B**) Time spent in active interaction with either a conspecific or non-social object in the 3-chamber social interaction test (3CSIT). Error bars show range, midlines of boxes show medians. (**C**) Results of the 2-way ANOVA with Dunnets multiple comparison *post-hoc* test, n = 11–12/group).

DOI: https://doi.org/10.7554/eLife.33070.003

**Table 1.** Meaningful pairwise comparisons for gene expression changes (nine unique comparisons).

| Group | Environmental stimulation | Compared to |
|---|---|---|
| CON (n = 4) | Naïve (n = 4, representing eight animals, per group) | GF, exGF, CON-SI |
| GF (n = 4) | | CON, exGF, GF-SI |
| exGF (n = 4) | | CON, GF, exGF-SI |
| CON-SI (n = 8) | social interaction (SI) | CON, GF-SI, exGF-SI |
| GF-SI (n = 12) | | GF, CON-SI, exGF-SI |
| exGF-SI (n = 8) | | exGF, CON-SI, GF-SI |

DOI: https://doi.org/10.7554/eLife.33070.004

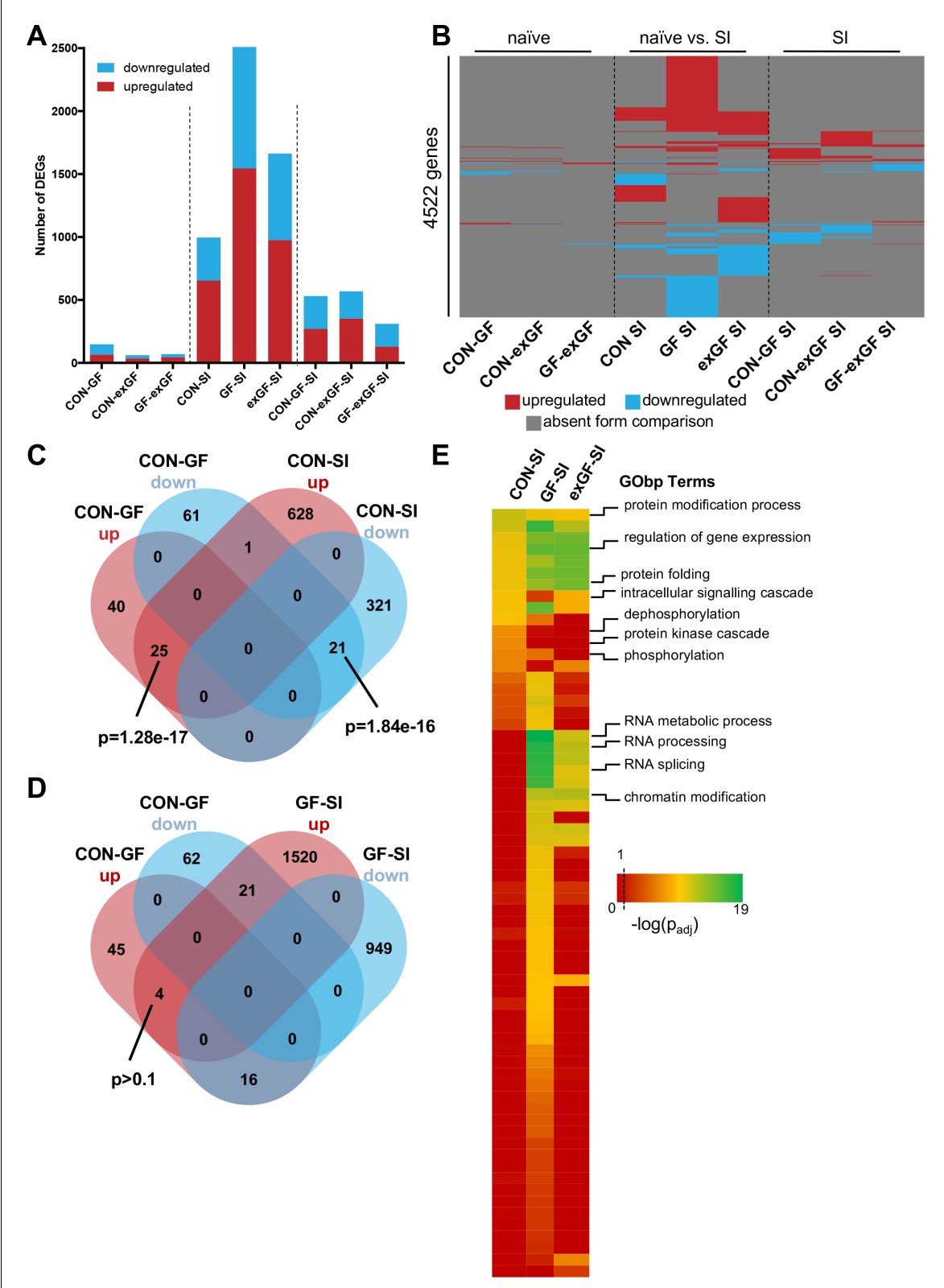

**Figure 2.** Social interaction induces diverging gene expression patterns in the amygdala. (A) Number of differentially expressed genes (DEGs) including both up- and downregulated genes for all meaningful pairwise comparisons (see **Table 1**). (B) Presence-absence plot for all DEGs in all groups. Each line represents one gene that is a DEG in at least one pairwise comparison. (C–D) Venn diagram of upregulated (up, red) and downregulated (down, blue) genes in CON-GF and CON-SI or GF-SI comparisons to test for activation of expression patterns in GF mice under naïve conditions that are

*Figure 2 continued on next page*

*Figure 2 continued*

similar to activation by social interaction at either CON or GF colonisation status. (**E**) Heatmap to visualize enriched biological functions in social interaction(SI)-induced (upregulated only) genes. FDR-adjusted p-values were calculated for all enriched ($p_{adj}$ <0.1) biological-function GO Terms (GObp), log-transformed (-$\log_{10}$) and colour-coded.

DOI: https://doi.org/10.7554/eLife.33070.005

The following figure supplements are available for figure 2:

**Figure supplement 1.** Additional data on gene expression analysis.

DOI: https://doi.org/10.7554/eLife.33070.006

**Figure supplement 2.** RT-qPCR validation of RNA-seq results.

DOI: https://doi.org/10.7554/eLife.33070.007

genes found in any of the comparisons (*Figure 2B*). As expected, there were multiple overlapping clusters between SI comparisons, demonstrating a core cluster of genes that are induced 1 hr after social novelty, independent of colonisation status. However, multiple clusters of genes showed differential regulation only in a particular group, suggesting differences in the transcriptomic response of the individual groups. We identified a cluster of genes that was differentially regulated in both GF naïve animals and upon social interaction stimulation in conventional animals (*Figure 2C*). This overlap was highly significant and made up large proportions of all genes up- or downregulated in naive GF animals compared to CON controls (38% and 26%, respectively). This cluster was not differentially regulated in GF animals upon stimulation by social interaction (*Figure 2D*). In fact, there was a counter directional overlap (e.g. genes upregulated in CON vs. GF but downregulated in GF upon stimulation). Together, these data suggest that several genes typically induced in the amygdala of conventional controls upon social interaction are already elevated in GF mice, pointing towards amygdalar hyperactivity in these animals (*Stilling et al., 2015*).

To identify pathways and biological functions that are induced in the amygdala by social interaction under the different colonisation conditions, we next analysed the dataset for functional enrichment. To this end, we tested upregulated genes in each group for enrichment of biological functions using the gene ontology (GO) database and compared resulting p-values (*Figure 2E*, *Supplementary file 2*). We found a wide-range of biological functions to be enriched in the three conditions. While overall the groups differed substantially, we identified one cluster that showed good agreement between comparisons. For all three groups significant enrichment was found for processes such as protein modification and folding as well as regulation of gene expression. This is in good agreement with the fact that there is a core cluster of genes overlapping (see *Figure 2B*). However, large differences were seen in processes that are associated with intracellular signalling such as protein phosphorylation and dephosphorylation, which was enriched in the CON-SI comparison, but not in GF-SI or exGF-SI. The second striking difference was dramatic enrichment of processes associated with RNA splicing, most prominently in the GF-SI but also the exGF-SI group (*Figure 2E*, *Supplementary file 3*). Using gene-set enrichment analysis (GSEA), which does not rely on cut-offs for p-value or fold-change, we find similar differences in functional enrichment (*Supplementary file 2*).

We also analysed the dataset for enrichment of genes associated with specific cellular pathways using the KEGG pathways database. As expected, in the CON-SI group the MAP kinase (MAPK) signalling pathway was the dominant pathway enriched due to social interaction treatment (*Figure 3A*, *Supplementary file 2*). This pathway, which is well established to be induced upon neuronal activity, was also enriched in GF-SI and exGI-SI groups, albeit to a much lesser degree. In fact, under more stringent analysis parameters enrichment failed to reach significance in these groups (*Figure 2—figure supplement 1A*). For the GF-SI group we found a highly significant enrichment of genes associated with the spliceosome that was also significantly enriched, to a lesser extent, in the exGF-SI group and almost completely absent from the CON-SI group (*Figure 3A*). To elucidate the differences in gene regulation in response to social interaction between CON and GF mice we analysed genes exclusively induced in either of the groups for further functional enrichment. In line with GO Term and KEGG pathway analysis we found that DEGs exclusively upregulated in CON-SI mice, were highly enriched in genes involved in intracellular signalling, especially the MAPK pathway, while DEGs exclusively upregulated in GF-SI mice were strongly enriched in genes associated with RNA processing, that is splicing (*Figure 3B*, *Supplementary file 2*). The core of genes that are induced

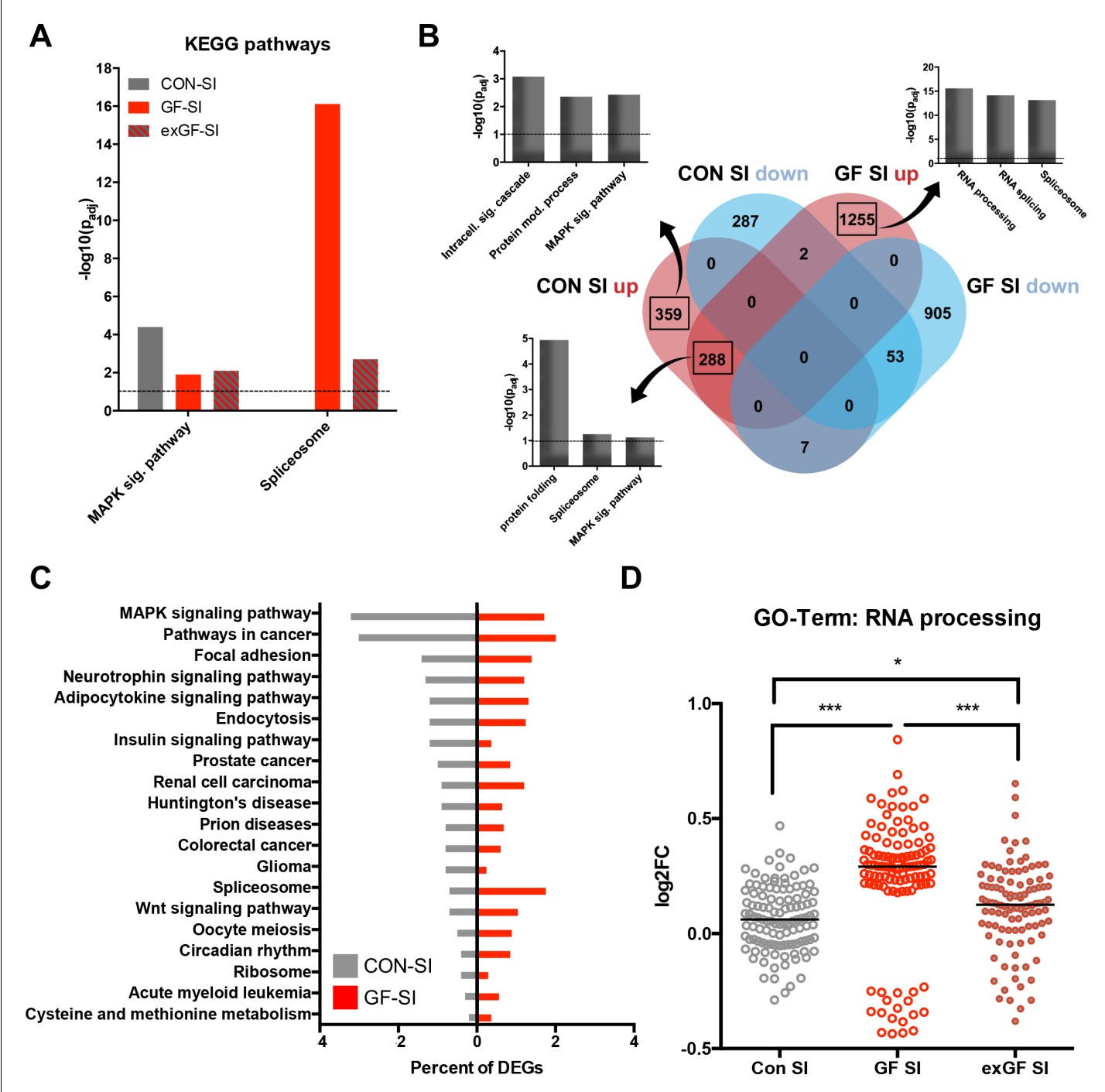

**Figure 3.** Loss of normal induction of MAP-K pathway is accompanied by upregulation of splicing pathways. (A) Comparison of enrichment of MAP-K pathway-associated genes and spliceosome-related genes among DEGs between groups. Dotted line: significance level (p=0.1) (B) Functional enrichment analysis of overlapping and uniquely regulated subsets of DEGs between CON-SI and GF-SI comparisons showed shared and different processes transcriptionally activated in the amygdala in response to social stimulation (C) Percentage of DEGs falling into the TOP20 enriched KEGG pathways for CON-SI and GF-SI. (D) Log$_2$(fold change) (Log2FC) values plotted for each of the 112 DEGs in the GF-SI comparison that was associated with GO-Term 'RNA processing'. black line: median, non-parametric Friedman test F = 65.5, p<1.0e-4; ***p<1.0e-4, *p<0.05, Dunn's multiple comparisons, n = 112/group.

DOI: https://doi.org/10.7554/eLife.33070.008

independent of colonisation status is characterised by enrichment of genes involved in protein folding and, to a lesser degree, both, spliceosome and MAPK signalling pathway. Genes falling into these two KEGG categories made up relatively large proportions of all DEGs in the CON-SI and the GF-SI group, respectively (*Figure 3C*, *Supplementary file 2*). Genes associated with the GO Term 'RNA processing' made up 4.4% of all DEGs in the GF-SI group (111/2510), which represents 15% of all mouse genes in this category (111/741). Comparing response-induced fold changes of these 111 genes between colonisation statuses, further highlighted that expression of this group of genes is highly distinctive in GF-SI animals (*Figure 3D*). This unique pattern was also evident when we plotted a comparison of these genes, where the individual gene identity was maintained across groups and where mean expression values for the remaining 630, not statistically significant mouse genes in the 'RNA processing' GO category for each group were included (*Figure 2—figure supplement 1B*).

Stimulus-induced upregulation of a number of well-established immediate early genes, that show rapid and reproducible upregulation in response to neuronal activity, including activity induced by environmental novelty in the hippocampus, was also seen in the amygdala of conventional animals after social interaction (*Figure 2—figure supplement 1C*). In line with reduced activation of the MAPK pathway in GF-SI mice, we find several of these genes, including *Egr3*, *Fos*, and *Ier5* not to be upregulated in this group, possibly due to elevated expression levels at baseline (*Supplementary file 1*).

Together these data suggest that GF animals show a unique response signature toward environmental stimulation by social interaction. The transcriptional response is characterised by a marked upregulation of the genes involved in RNA processing and the splicing machinery, accompanied by an arrested upregulation of typical response genes, especially in those involved in MAPK signalling. Analysis of differential gene expression between naïve GF animals and controls also revealed that this lack of induction may be due to upregulation of several genes involved in these processes already at baseline. All analyses on GF animals colonised with a conventional microbiota at weaning (exGF) presented so far show results with features of both, CON and GF groups.

## Social interaction leads to a strong increase in alternative splicing in GF and exGF animals

Since 'RNA processing' and spliceosome-associated pathways were highly enriched in the amygdala of GF animals and also exGF animals, we next analysed our dataset for functional consequences of this upregulation by searching for alternative splicing events, that is differentials spliced genes (DSGs). Under naïve conditions alternative splicing was highest when comparing CON and GF animals (*Figure 4A*, *Supplementary file 4*). These genes were highly enriched in several functional categories involved in neuronal function, such as long-term potentiation ($p_{adj}$ = 2.3e-6) and synaptic transmission ($p_{adj}$ = 1.2e-10) (*Supplementary file 5*). Similar enrichment, although to a lesser degree was found when comparing GF and exGF animals, while no significant functional enrichment was found for the relatively few DSGs in the CON vs. exGF comparison. High congruence in alternative splicing between CON and exGF mice was also evident from the high degree of overlap between DSGs that distinguished both groups from the GF group (*Figure 4B*). Together these data are in agreement with phenotypic similarities between CON and exGF animals, that is reversibility of the GF phenotype by colonisation at weaning.

In line with diverging transcriptomes between groups due to differential gene expression after stimulation by social novelty, the transcriptional landscapes changed substantially as a result of alternative splicing. While conventionally colonised controls exposed to social interaction showed comparatively little alternative exon usage, we found a dramatically increased number of DSGs in the GF-SI group, and noticed an even higher number of SI-induced DSGs in exGF animals. This was surprising since enrichment of splicing-associated genes among DEGs was highest in the GF-SI comparison, albeit significantly present also in the exGF-SI group.

As with the previous finding for DEGs, DSGs of the three comparisons also shared a 'core' set of genes that were alternatively spliced in response to social interaction, independent of colonisation status (*Figure 4C*). This core was slightly enriched in genes associated with functional categories that are involved in neuronal function such as long-term potentiation ($p_{adj}$=1.5e-02) and synaptic transmission ($p_{adj}$=1.9e-4). In fact, the high degree of overlap between all three groups together with the high number of DSGs in GF-SI and exGF-SI groups suggests that, in addition to the conventional response, these two groups show supplementary cellular responses towards activation of the

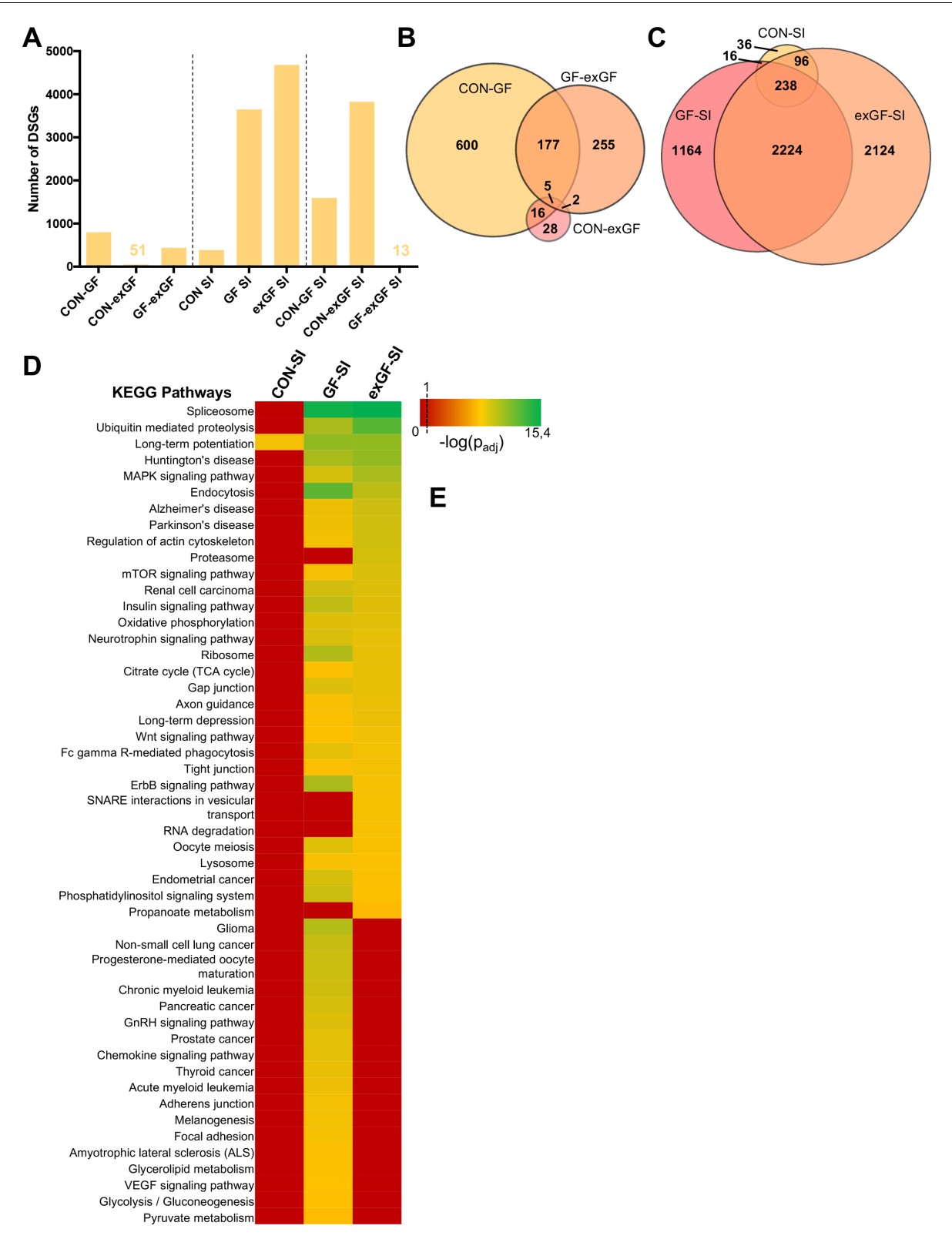

**Figure 4.** Social interaction leads to a strong increase in alternative splicing in GF and exGF animals. (A) Number of differentially spliced genes (DSGs) for all meaningful pairwise comparisons (see *Table 1*). (B–C) Venn diagrams showing overlapping DSGs for naïve (B) and social interaction (SI, (C)) comparisons. After stimulation differences become evident. (D) Heatmap to visualize enriched pathways in SI-induced differentially spliced genes. FDR-adjusted p-values were calculated for all enriched ($p_{adj}$ <0.1) KEGG pathways, log-transformed ($-\log_{10}$) and colour-coded.

*Figure 4 continued on next page*

*Figure 4 continued*

DOI: https://doi.org/10.7554/eLife.33070.009

amygdala by social interaction. Interestingly, this additional response is even more pronounced in the exGF-SI group and is characterised by a high number of DSGs highly enriched in genes involved in mainly four principal functional categories: splicing/RNA processing, protein turnover, neuronal functions, and intracellular signalling pathways (*Figure 4D*, *Supplementary file 6*). The fact that a highly enriched proportion of the alternatively spliced genes themselves were also members of the splicing machinery suggests a self-regulating mechanism of gene expression in these two groups.

## Behavioural performance correlates with expression of RNA processing genes in GF mice

Given the high behavioural variability in active social interaction time specifically in the GF-SI group, together with the divergence of the transcriptional landscape in the amygdala one hour after exposure, we hypothesised that gene expression may be correlated with behavioural performance in individual mice. To test this hypothesis, we ranked all expressed genes in our dataset by correlation (Pearson correlation coefficient) between expression level and time spent in active social interaction (*Supplementary file 7*). In order to identify functions that are associated with positively or negatively correlated genes we used the *Gene Set Enrichment Analysis* (GSEA) algorithm that assigns one or multiple functions to each gene in the list based on the GO Term or KEGG Pathway databases (*Subramanian et al., 2005*). The algorithm then calculates an enrichment score for each function based on the rank of genes associated with this function (*Figure 5B,C*). Excitingly, genes whose expression was positively correlated with social interaction time were significantly associated with splicing, along with protein turnover (*Figure 5B–D*). There was no significant gene set enrichment for negative correlations between sociability behaviour and gene expression. These data suggest direct involvement of splicing and protein turnover pathways in the amygdala with behavioural performance. This result was confirmed, when we compared gene expression of the six germ-free animals with the highest social behaviour performance (GF-SI[high]) with those six animals with lowest social behaviour performance (GF-SI[low]). We found three genes that were significantly higher expressed in GF-SI[high] animals, namely *DnaJb6* (encoding a brain-enriched heat-shock family protein with chaperone function), *Cirbp* (encoding a stimulus-inducible, brain-enriched, nuclear RNA-binding protein involved in mRNA stabilization), and *D030028A08Rik* (a long-noncoding RNA with yet unknown function). These three genes also show a statistically significant correlation with behavioural performance in the GF-SI group (*Figure 5E*).

## Discussion

The amygdala is a key node in the emotional processing network, responsible for processing social stimuli and fear-related cues (*Allsop et al., 2014*; *Amaral, 2003*; *Kliemann et al., 2012*; *Noonan et al., 2014*; *Phelps and LeDoux, 2005*; *Sabatini et al., 2007*; *Sallet et al., 2011*; *Sliwa and Freiwald, 2017*). A rapidly developing literature increasingly implicates the microbiome in host brain function and behaviour, especially in these emotion processing networks (*Crumeyrolle-Arias et al., 2014*; *Luczynski et al., 2016a*; *Clarke et al., 2013*; *Diaz Heijtz et al., 2011*; *Neufeld et al., 2011*; *Arseneault-Bréard et al., 2012*; *Bercik et al., 2010*; *Gilbert et al., 2013*). Social behaviour appears to be among the behaviours most intimately connected to a functional microbiome (*Arentsen et al., 2015*; *Desbonnet et al., 2014*; *Gacias et al., 2016*; *Lewin-Epstein et al., 2017*; *Theis et al., 2013*; *Tung et al., 2015*; *Hsiao et al., 2013*; *Buffington et al., 2016*; *Arentsen et al., 2017*; *Koch and Schmid-Hempel, 2011*). However, the mechanistic underpinnings of this influence are only beginning to be resolved. Here we show, for what is to our knowledge the first time, that absence of the microbiome results in dysregulation of unique transcriptional-response pathways in the amygdala.

In line with previous reports, we show that microbes are necessary for development of appropriate sociability behaviour (*Arentsen et al., 2015*; *Desbonnet et al., 2014*; *Buffington et al., 2016*). In agreement with these studies we show reduced sociability behaviour of GF mice in the three-

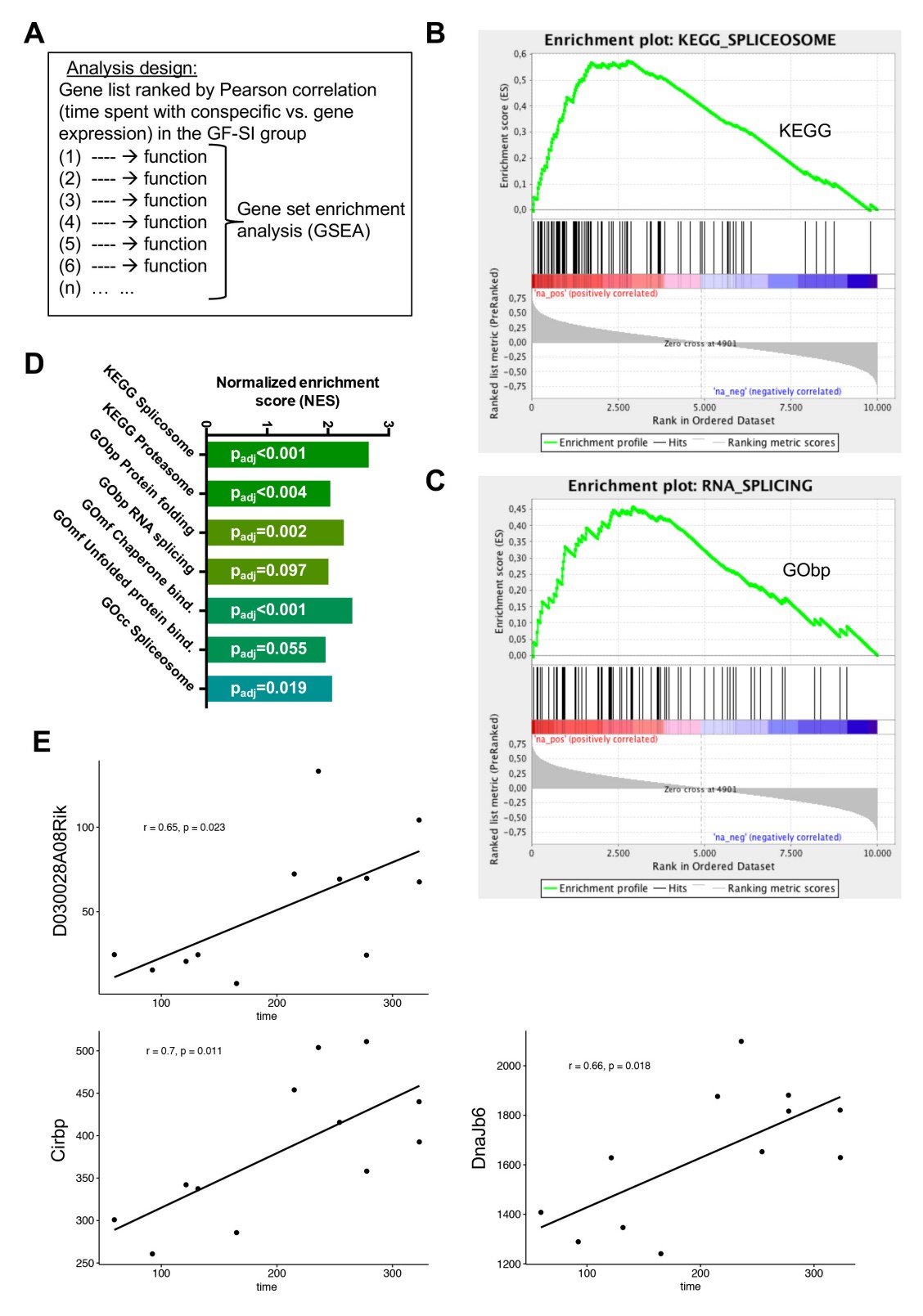

**Figure 5.** Behavioural performance correlates with expression of RNA processing genes in GF mice. (A) Design of analysis. A gene list, ranked by Pearson correlation between expression level of a given gene and behavioural performance for each individual mouse, was used for gene-set enrichment analysis (GSEA). The algorithm assigns one or multiple functions to each gene in the list based on the GO-Term or KEGG Pathway databases, then calculates an enrichment score for each function based on the rank of genes associated with this function (B–C) GSEA-generated

*Figure 5 continued on next page*

*Figure 5 continued*
enrichment plots for two selected significantly enriched functions (KEGG pathway 'Spliceosome', GO-Term 'RNA splicing'). (**D**) Normalized GSEA enrichment scores for all significantly enriched functions or pathways found in this analysis. (**E**) Gene-expression analysis within the GF-SI group (GF-SI$^{high}$ vs GF-SI$^{low}$, n = 6). Shown are expression values for three significantly differentially regulated genes plotted against the animals time spent in active social interaction. r: Pearson correlation coefficient.
DOI: https://doi.org/10.7554/eLife.33070.010

chamber social interaction task, rescued by post-weaning colonisation with a conventional micro-biome. However, we do not observe a lack of preference of the conspecific mouse over an inanimate novel object in GF mice. Interestingly, while on average the group of GF mice spent significantly less time interacting with a conspecific than conventionally raised animals, some individual mice of this group performed at control level. This finding suggests, that the underlying networks controlling sociability behaviour are subject to a dynamic regulation, possibly associated with differences in intracellular and extracellular neuronal signalling pathways due to subtle differences accumulating during individual development. However, the fact that exGF mice, colonised with a conventional microbiota at weaning age, spent an intermediate amount of time in social interaction with more control-like variability within the group, argues that this development is highly susceptible to influence by symbiotic signals from the microbiota. Also in several analyses of gene expression regulation in response to social interaction exGF mice resemble an intermediate phenotype, bearing features of both CON and GF mice. As such, exGF mice show high enrichment of genes involved in 'regulation of gene expression' (most prominently enriched among CON-SI upregulated genes) as well as 'RNA processing' (most prominently enriched among GF-SI upregulated genes). This intermediate gene expression phenotype is also evident from the visualized functional enrichment using colour-coded functional GO-Terms (*Figure 2E*).

To characterize real transcriptome-wide changes in gene expression, in this study we used stranded, ribodepleted as opposed to poly-A enriched libraries for RNA-sequencing. In result, we describe dynamic regulation of several previously undescribed pathways in response to environmental stimulation. As such, we see regulation of RNA-processing non-coding RNAs, several of which are found in subnuclear Cajal bodies, that are particularly prominent in neurons - especially, when transcriptionally active - and are crucially involved in splicing regulation (*Wang et al., 2016*; *Lafarga et al., 2017*). Our RNA-seq experiment thus offers exclusive and comprehensive insight into gene regulation in response to a social stimulus in the amygdala. Interestingly, induction of gene expression in this brain region shows similarities with the hippocampal transcriptional response to environmental novelty (*Stilling et al., 2014c*). This overlap is very likely due to a ubiquitous, though highly specific transcriptional response in neurons towards neuronal activity, which includes induction of several well established immediate early genes such as *Fos* or *Arc*, the MAP-K pathway, and neurotrophic signalling via *Bdnf*. Moreover, we find upregulation of complement components, which have lately been established to be necessary for synaptic rearrangements and plasticity upon neuronal activity (*Schafer et al., 2012*; *Stephan et al., 2012*; *Stevens et al., 2007*). Interestingly, innate immune system genes together with neuronal activity-dependent genes have recently been shown to be dysregulated in autism (*Gupta et al., 2014*). Induction of complement genes upon social interaction was not seen in GF mice, possibly due to upregulation of *C1q* already under naïve conditions as compared to CON mice. Indeed, we find that a highly significant share of genes upregulated upon social interaction stimulation in CON mice is already upregulated in naïve GF mice. In perfect agreement with previous reports (*Hoban et al., 2018*; *Stilling et al., 2015*), this finding suggests baseline hyperactivity of neurons in the amygdala in GF mice. Together, these findings confirm that the expected response of the control group serves as a positive control and provide internal validation that the methodology of this study is able to detect relevant changes in transcriptional regulation between groups.

Our data further shows that altered splicing activity is a normal process in neurons of the amygdala in response to social interaction in conventionally raised mice. Indeed, it is now well established, that neuronal activity induces alternative splicing patterns (*Ding et al., 2017*; *Hermey et al., 2017*; *Iijima et al., 2016*; *Schor et al., 2009*), a dysfunction of which has been associated with changes seen in autism and autistic-like behaviour in mice (*Quesnel-Vallières et al., 2016*). Surprisingly, the regulation of genes involved in splicing, including Cajal body-associated genes, as well as alternative

splicing activity is extremely exaggerated in GF mice, which possibly reflects a compensatory mechanism for already elevated activity-induced signalling at baseline. The finding that increased splicing activity is even more amplified in exGF mice, which display a largely normalized behavioural response, together with a positive correlation between expression of splicing-associated genes and behavioural performance suggests that in fact upregulation of the splicing machinery supports an adequate amygdalar response towards a social interaction environmental stimulus. In fact, previous research shows that post-weaning colonization (exGF) does not rescue impairments in social cognition, seen in GF mice (*Desbonnet et al., 2014*). This is likely a reflection of an incomplete rescue of the molecular underpinnings of social behaviour investigated here. Although beyond the scope of the current study and technically challenging, future investigations should focus on untangling the network and pathways that drive the observed RNA processing changes and derive molecular consequences from altered mRNA/protein amino acid sequences.

In summary, our data is fully congruent with and offers a molecular basis for previous data on alterations in social cognition (*Desbonnet et al., 2014*), amygdala volume and dendrite complexity (*Luczynski et al., 2016a*), and increased transcription of activity-associated gene expression in the amygdala (*Hoban et al., 2018*; *Stilling et al., 2015*) in microbiome-deficient mice. We here show for what is to our knowledge the first time that the microbiota is necessary for regulation of core biological processes on the molecular and cellular level in the brain, which makes a strong case for a causal involvement of the microbiota in the molecular mechanisms leading to the observed impairments in sociability behaviour and the aetiology of neurodevelopmental diseases, which opens the possibility for new therapeutic strategies.

## Materials and methods

### Animals

Male $F_1$-generation offspring from germ-free (GF) and conventionally-raised (CON) Swiss Webster breeding pairs previously obtained from Taconic (Germantown, New York, USA) were used in all experiments as previously described (*Desbonnet et al., 2014*; *Stilling et al., 2015*; *Clarke et al., 2013*; *O'Tuathaigh et al., 2007*). GF mice were housed in groups of two-four per cage in flexible-film gnotobiotic isolators at a 12 hr light/dark cycle. Ex-germ-free (exGF) mice were removed from the GF unit after weaning on postnatal day p21, and housed on CON-used bedding next to CON mice in the standard animal unit to allow colonization of microbes present in the facility environment. CON mice were similarly housed two–five per cage under controlled conditions (temperature 20–21°C, 55–60% humidity) on the same 12 hr light/dark cycle. All groups received the same autoclaved, pelleted diet (Special Diet Services, product code 801010). Age at tissue extraction for all groups and experiments was 10 weeks.

### 3-chamber social interaction test (3CSIT)

The 3CSIT was performed as described previously (*Desbonnet et al., 2014*; *Stilling et al., 2015*). In brief, mice were habituated to the test room for half an hour and then habituated to a white plastic arena (40 × 20×20 cm), divided into three chambers by separators with small circular openings and lined with fresh bedding, for 10 min. The left and right chamber contained empty wire-mesh cages. These were then used to display an age- and sex-matched conventionally-raised mouse (chamber 1) or a mouse-sized and –coloured (white) non-social object during the test phase (porcelain egg cup). Exploration of the three chambers by the test mouse was recorded on video for 10 min and time spent in active interaction with the conspecific or object was measured by an experimenter, blinded to colonization status of the test mice. Group size for behaviour was n = 12 for GF-SI mice and n = 11 per group for CON-SI and exGF-SI groups, after removing outliers.

### RNA extraction and sequencing

The amygdala from the left brain hemisphere was rapidly dissected on ice from fresh brain tissue as adapted from (*Zapala et al., 2005*), stored in RNA*later* RNA Stabilization Reagent (Qiagen, Netherlands) at 4°C for 24 hr and then transferred to −80°C. Total RNA was extracted using the *mir*Vana™ miRNA Isolation kit (Ambion/life technologies) and DNase treated (Turbo DNA-free, Ambion/life technologies) according to the manufacturers recommendations For each group (CON, GF, exGF,

CON-SI, GF-SI, exGF-SI), 8–12 animals were used (see *Table 1* for details). RNA concentration and quality were determined using a Nanodrop 1000 (Thermo Scientific) and a Bioanalyzer (Agilent) was used to measure RNA integrity. After this, for all naïve groups equal amounts of RNA from two animals were then pooled to yield four samples per group. Therefore, for naïve groups, each sample (technical replicate) analysed by RNA-seq represents the average of two biological replicates.

Ribodepletion and library preparation was performed by Vertis Biotechnology (Freising, Germany). Sequencing as well as Fastq-file generation was done by Beckman Coulter Genomics service (Danvers, MA, USA). Stranded, paired-end reads of 2 × 100 bp were produced on an Illumina HiSeq2500 sequencer. Details on RNA sample quality and sequencing quality control are given in *Supplementary file 8*.

## Molecular validation by quantitative real-time PCR (qRT-PCR) for selected differentially regulated genes

To validate differential expression results by the RNAseq pipeline, we performed qRT-PCR analysis of all individual RNA samples for the six groups for 18 selected genes (Figure 2 - figure supplement 2). 61% of comparisons showed matching results between the two methods. qRT-PCR was performed as previously described (*Stilling et al., 2015*). In brief, qRT-PCR of 3 technical replicates was done for each biological sample on a LightCycler 480 system (Roche LifeScience) and analysed using the $\Delta\Delta C_t$ – method. Two-way ANOVA with multiple-testing correction (Tukey *post-hoc* for effect of colonisation status; Sidak *post-hoc* for effect of social interaction stimulation), was used to test for statistical significance between groups for each gene. Significance level was: $p_{adj} < 0.05$.

## Bioinformatic analysis pipeline

### Quality control and mapping to reference genome

Fastq-format reads were quality filtered and trimmed using Trimmomatic (v0.32, RRID:SCR_011848) (*Bolger et al., 2014*) with the following non-default parameters: *AVGQUAL*: 20; *SLIDINGWINDOW*: 4:20; *LEADING*: 10; *TRAILING*: 10; *MINLEN*: 60. Alignment to the mouse reference genome (GRCm38.p3) was achieved using the STAR aligner (v2.4.0f1) (*Dobin et al., 2013*) with default options and an index compiled with gene models retrieved from the Ensembl database (release 78).

### Differential gene expression and functional enrichment analyses

Ensembl database release 78 gene models were also used for counting mapped reads per gene using HTSeq-Count (v0.6.0, RRID:SCR_011867) (*Anders et al., 2015*) with the following non-default parameters: -s: no; -r: pos; -q –f bam –m intersection-nonempty. Differential gene expression was calculated for pairwise comparisons using the DESeq2 R-package (v1.6.2) (*Anders and Huber, 2010*; *Love et al., 2014*) with default parameters. Genes with an FDR-adjusted p-value≤0.1 were considered differentially regulated. The data discussed in this publication have been deposited in NCBI's Gene Expression Omnibus (*Edgar et al., 2002*) and are accessible through GEO Series accession number GSE114702 (https://www.ncbi.nlm.nih.gov/geo/query/acc.cgi?acc=GSE114702).

Differentially expressed genes were further analysed for functional enrichment using the DAVID Bioinformatic Resources (v6.7) (*Huang et al., 2009a*; *Huang et al., 2009b*) and Gene Set Enrichment Analysis (GSEA) (v2.0.14, RRID:SCR_003199) (*Subramanian et al., 2007*).

### Differential exon usage analysis

For detection of differential exon usage the DEXSeq R-package was used (v1.12.2, RRID:SCR_012823) (*Anders et al., 2012*) with default parameters. FDR-corrected p-value significance level was set to 0.1. Ensemble database release 78 was used to provide exon models.

## Statistics

Graphpad Prism (v6.0h, RRID:SCR_002798) was used for statistical hypothesis testing of behavioural analysis (*Figure 1*) and expression of RNA processing genes (*Figure 3D*). Before hypothesis testing on behavioural data, normal distribution was confirmed by D'Agostino and Pearson omnibus normality, Shapiro-Wilk normality, and KS normality tests (p<0.05 for all three tests). Sample size of n = 12 was based on a power calculation with an expected effect size of f = 0.4, a significance level of α = 0.05 and a power of 0.8. A Grubbs test for outliers (p<0.05) identified one outlier each in CON-

SI and exGF-SI groups. One-way ANOVA with Tukey's multiple comparisons post-hoc test was run to compare social interaction between the groups.

For hypothesis testing of expression of RNA processing genes non-parametric Friedman testing was performed with Dunn's multiple comparison post-hoc test to compare expression levels between groups.

Statistical significance for overlaps of differentially expressed genes in pairwise comparisons (*Figure 2C,D*) was computed using a publicly available web service (http://nemates.org/MA/progs/overlap_stats.html), which is based on the hypergeometric distribution and Fischer's exact test.

Heatmaps for functional enrichment (*Figure 2E*, *Figure 4D*): FDR-adjusted p-values were calculated for all enriched ($p_{adj}$ <0.1) biological-function GO Terms using DAVID Bioinformatic Resources (v6.7) (see above) and log-transformed (-$\log_{10}$) and colour-coded using Microsoft Excel (v15).

Pearson correlation coefficient for correlation between behavioural performance and gene expression level in individual mice was calculated using Microsoft Excel (v15).

## Acknowledgements

The authors wish to thank Mr. Patrick Fitzgerald, and Ms. Frances O'Brien for technical assistance with animal husbandry, tissue extraction, and RNA extraction. The APC Microbiome Institute is a research centre funded by Science Foundation Ireland (SFI), through the Irish Government's National Development Plan (Grant Number 12/RC/2273). RMS was supported by the Irish Research Council (IRC) through a Government of Ireland Postdoctoral Fellowship (Grant Number GOIPD/2014/355). MJC and FJR are also supported by Science Foundation Ireland (Grant Number 11/SIRG/B2162.) T G D and J F C are also supported by the Irish Health Research Board, the Dept. of Agriculture, Food and Fisheries and Forestry and Enterprise Ireland. GC is supported by a NARSAD Young Investigator Grant from the Brain and Behavior Research Foundation (Grant Number 20771).

## Additional information

### Competing interests

Fergus Shanahan: principal investigator in the APC Microbiome Institute, University College Cork. Gerard Clarke, Marcus J Claesson: faculty member or funded investigator of the APC Microbiome Institute. The APC Microbiome Institute has conducted research funded by Pfizer, GlaxoSmithKline, Proctor & Gamble, Mead Johnson, Suntory Wellness, and Cremo. Timothy G Dinan: principal investigator in the APC Microbiome Institute, University College Cork. Has been an invited speaker at meetings organized by Servier, Lundbeck, Janssen, and AstraZeneca. John F Cryan: principal investigator in the APC Microbiome Institute, University College Cork. Has been an invited speaker at meetings organized by Mead Johnson, Yakult, Alkermes, and Janssen. The other authors declare that no competing interests exist.

### Funding

| Funder | Grant reference number | Author |
| --- | --- | --- |
| Science Foundation Ireland | 12/RC/2273 | Fergus Shanahan<br>Gerard Clarke<br>Marcus J Claesson<br>Timothy G Dinan<br>John F Cryan |
| Irish Research Council | GOIPD/2014/355 | Roman M Stilling<br>John F Cryan |
| Irish Health Board | | Timothy G Dinan<br>John F Cryan |
| National Alliance for Research on Schizophrenia and Depression | 20771 | Gerard Clarke |

The funders had no role in study design, data collection and interpretation, or the decision to submit the work for publication.

## Author contributions

Roman M Stilling, Conceptualization, Data curation, Formal analysis, Funding acquisition, Investigation, Visualization, Methodology, Writing—original draft, Project administration, Writing—review and editing; Gerard M Moloney, Alan E Hoban, Formal analysis, Investigation, Methodology, Writing—review and editing; Feargal J Ryan, Data curation, Software, Formal analysis, Visualization, Methodology, Writing—review and editing; Thomaz FS Bastiaanssen, Software, Formal analysis, Methodology, Writing—review and editing; Fergus Shanahan, Gerard Clarke, Resources, Supervision, Funding acquisition, Writing—review and editing; Marcus J Claesson, Resources, Software, Supervision, Funding acquisition, Methodology, Writing—review and editing; Timothy G Dinan, Conceptualization, Resources, Supervision, Funding acquisition, Writing—review and editing; John F Cryan, Conceptualization, Resources, Supervision, Funding acquisition, Investigation, Methodology, Writing—original draft, Project administration, Writing—review and editing

## Author ORCIDs

Roman M Stilling https://orcid.org/0000-0001-7637-5851
Gerard M Moloney http://orcid.org/0000-0002-3672-1390
Feargal J Ryan https://orcid.org/0000-0002-1565-4598
Thomaz FS Bastiaanssen http://orcid.org/0000-0001-6891-734X
Gerard Clarke http://orcid.org/0000-0001-9771-3979
Marcus J Claesson http://orcid.org/0000-0002-5712-0623
John F Cryan http://orcid.org/0000-0001-5887-2723

## Ethics

Animal experimentation: This study was performed in strict accordance with the recommendations provided by Laboratory Animal Science and Training (LAST) Ireland. All of the animals were handled according to institutional protocols approved by the Animal Ethics Experimentation Committee (AEEC) of University College Cork (#2015/014) and the Health Products Regulatory Authority (HPRA) Ireland (#AE19130/P023). Every effort was made to minimize suffering and animals were killed humanely.

## Decision letter and Author response

Decision letter https://doi.org/10.7554/eLife.33070.024
Author response https://doi.org/10.7554/eLife.33070.025

# Additional files

## Supplementary files

• Supplementary file 1. Differentially expressed genes
DOI: https://doi.org/10.7554/eLife.33070.011

• Supplementary file 2. Functional enrichment DEGs
DOI: https://doi.org/10.7554/eLife.33070.012

• Supplementary file 3. DEGs GO-heatmap
DOI: https://doi.org/10.7554/eLife.33070.013

• Supplementary file 4. DSGs
DOI: https://doi.org/10.7554/eLife.33070.014

• Supplementary file 5. Functional enrichment DSGs
DOI: https://doi.org/10.7554/eLife.33070.015

• Supplementary file 6. DSGs KEGG-heatmap
DOI: https://doi.org/10.7554/eLife.33070.016

• Supplementary file 7. Correlation gene expression with behaviour

DOI: https://doi.org/10.7554/eLife.33070.017

• Supplementary file 8. Details on RNA sample quality and sequencing quality control
DOI: https://doi.org/10.7554/eLife.33070.018

• Supplementary file 9. Raw FastQC quality control files for sequencing data
DOI: https://doi.org/10.7554/eLife.33070.019

• Transparent reporting form
DOI: https://doi.org/10.7554/eLife.33070.020

## Data availability

The data discussed in this publication have been deposited in NCBI's Gene Expression Omnibus [86] and are accessible through GEO Series accession number GSE114702 https://www.ncbi.nlm.nih.gov/geo/query/acc.cgi?acc=GSE114702

The following dataset was generated:

| Author(s) | Year | Dataset title | Dataset URL | Database, license, and accessibility information |
|---|---|---|---|---|
| Stilling RM, Moloney GM, Ryan FJ, Hoban AE, Bastiaanssen TF, Shanahan F, Clarke G, Claesson MJ, Dinan TG, Cryan JF | 2018 | Social interaction-induced activation of RNA splicing in the amygdala of microbiome-deficient mice | https://www.ncbi.nlm.nih.gov/geo/query/acc.cgi?acc=GSE114702 | Publicly available at the NCBI Gene Expression Omnibus (accession no: GSE114702) |

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
