## [Decision Letter]

Thank you for sending your article entitled "Social Interaction-induced Activation of RNA Splicing in the Amygdala of Microbiome-Deficient Mice" for peer review at *eLife*. Your article has been evaluated by three peer reviewers, and the evaluation has being overseen by a guest Reviewing Editor and Wendy Garrett as the Senior Editor. The reviewers have opted to remain anonymous.

Summary:

The manuscript by Stilling et al. uses transcriptomic analyses to identify differential changes in gene expression in amygdala of conventionally colonized vs germ-free mice after social stimulation. The study premise is of significant interest; however, there are concerns regarding the quality and robustness of the dataset that the reviewers hope the authors are able to address and resolve. Specifically, there is concern that the changes in gene expression are moderate relative to the large variation between replicates. Additional methodological controls are absolutely necessary and several textual revisions are needed to support the primary findings:

Essential revisions:

1) Molecular validation of key differentially expressed genes identified by transcriptomic analysis. Reviewers raised the concern that the manuscript is highly descriptive and lacks qPCR validation of key differentially expressed genes.

2) Examination of within-group gene expression changes relative to behavioral performance (e.g., comparison between germ-free mice that demonstrated social interaction behaviours similar to controls and those that clearly showed a deficit).

3) Examination of data relative to effect size of differential expression.

4) Comment on disagreement between behavioral phenotype reported in this study as compared to previous report in Desbonnet et al., 2014. The reviewers raised the concern that the germ free mice do not show impairment in social behavior in this study, as compared to previous findings from the same group.

5) Discussion of specific splicing factors in the context of existing literature and broader implications, and comment on whether changes in any splicing factors can explain the changes in exon usage.

6) Additional methodological details of quality controls for sequencing, such as RIN number of the RNA samples, sequencing depth, mapping rate, the number of exonic reads, 3' bias, etc. are needed to evaluate the quality of the data. Also there is question as to whether the analytical software used considers whether alternative splicing occurs using exon junction reads. Consideration of how many of the reported exons are alternative exons based on existing annotations is needed.

7) More nuanced discussion of the implications of findings. The reviewers request a more focused discussion of conclusions that can be drawn from the provided data.

*Reviewer #1:*

Stilling et al. employed transcriptomics to determine the influence of the gut microbiome on social-interaction-induced changes in gene expression in the amygdala. By comparing naïve vs. socially stimulated mice, both control and germ-free (GF) mice, the authors found that social interaction triggers a differential change in gene expression in GF mice compared to control mice. Changes in the transcriptional landscape and GF social deficits were partially reversed by colonization at weaning. The authors identified significant overlap in a core population of genes that were differentially expressed in both naïve GF animals and socially stimulated controls. Among these, genes encoding elements of the MAPK pathway, which regulates neural plasticity, were upregulated in naïve GF animals, but only after social stimulation in conventionally colonized controls. Moreover, social stimulation markedly upregulated genes involved in RNA processing and splicing in GF animals, which was not observed in either control or GF animals colonised at weaning. Finally, the authors determined that the extent of alternative splicing observed in the amygdala of an individual correlated with the severity of the behavioral deficit observed in the animal.

I find the paper by Stilling et al. to be an interesting piece of work. However, as it stands, the paper suffers from several deficiencies. First, the work is primarily descriptive and there is no molecular validation of the bioinformatic results. Second, and perhaps more problematic, the GF mice do not show impairment in social behavior. Briefly, GF mice spent more time interacting with a mouse vs. an inanimate object, which is inconsistent with previous findings from the same group. Finally, it is not clear why the study focuses exclusively on the amygdala. A more thorough study comparing gene expression pattern from brain regions involved in social behaviors will strengthen this study. My specific comments are below:

1) The data presented in Figures 1B and C are inconsistent with a previous study from this group (Desbonnet et al., 2014) in which GF animals preferred interaction with an empty cup over interaction with a mouse in the 3-chamber task. This inconsistency should be addressed and the data in panels B and C should be combined into a single panel with accompanying statistical data from both intra- and inter-group two-way ANOVAs.

2) The authors should validate the top 10-25 transcripts reported in Figure 2 by qPCR or Western blotting and provide these data in the figure. In the same line, MAPK pathway activity is readily accessible for analysis by Western blotting. Can the authors provide molecular validation of the MAPK pathway data-baseline upregulation in GF animals and post-interaction upregulation in controls?

3) It's unlikely that the alternative splicing events in GF animals are restricted to the amygdala. Other structures including those in the mesolimbic dopaminergic reward system, specifically the nucleus accumbens and ventral tegmental area, are emerging as regions critically involved in regulating social behavior. Did the authors perform similar transcriptomic analysis on tissue isolated from other brain regions following the interaction? This data would bolster the authors' claim that colonization-state dependent alterations in RNA splicing and processing exerts meaningful influence over social behavior.

4) The lengthy Discussion presents an interesting theory unifying neuroimmune interaction data with findings from the microbiome and behavior field; however, this is completely outside of the scope of the conclusions that can be drawn from the provided data and, in my view, the implications of this study should be toned down.

*Reviewer #2:*

This article explores the potential underlying molecular mechanisms of impaired social behaviour in Germ-Free (GF) mice. A key finding is that following an established social interaction test, GF animals exhibited an increased expression of genes encoding splicing factors in the amygdala, relative to controls. Although GF mice that were colonised with microbiota after weaning demonstrated similar molecular changes, they were greater than in GF mice. Overall the manuscript documents extremely important observations that not only lend further support for the involvement of the microbiome in sociability, but also provides essential, novel data that can be interrogated more extensively to decipher how microbial communities might influence host central function. This report is well written, the figures are excellent and the state-of-the-art methodologies and statistical analyses used were entirely appropriate. My comments are:

1) Not sure if comparisons with behaviour naïve and tested mice really add to the study, and perhaps can be moved to supplementary data?.…The work is not addressed in the Discussion anyway – what key information can the reader glean from these associations?

What really would have been interesting to see are comparisons between the GF mice that demonstrated social interaction behaviours similar to controls (>300 sec), with those that clearly showed a deficit (<200sec) in Figure 1B i.e. what are the molecular profiles within the GF-SI cohorts?.…The authors did mention variability amongst these animals, so are these variabilities reflected at the molecular level?.…Obviously a meaningful analysis might be under-powered, but even a cursory inspection would be valuable.

2) Subsection “Social interaction induces highly distinct gene expression patterns in the amygdala of germ-free mice”, first paragraph: results should be presented immediately, perhaps with an introductory sentence – the preceding detailed paragraph with references is too excessive – – this should be moved to Materials and methods or Introduction –.…but ideally, as mentioned above, these comparisons should be hidden in the shadows so the reader can focus on the main findings between CON-SI, GF-SI & exGF-SI!

*Reviewer #3:*

Stilling et al. hypothesized that social behavior can be modulated by host-associated microbiota. To test this hypothesis, the authors compared control mice, germ-free mice and GF mice colonized post weaning by RNA-seq analysis of amygdala upon exposure to social interaction. The authors found significant gene expression changes between groups. In particular, GF mice with SI exposure showed increased expression of splicing factors, as well as differential splicing of a large number of exons. While the hypothesis is potentially interesting, this reviewer is concerned whether the analysis has properly controlled and reproducible to make the claims. Overall, the work is also quite descriptive in general.

1) The authors judge the difference of two conditions by the number of differentially expressed genes. This is problematic because the number of differentially expressed genes is largely affected by the number of replicates and sequence depth, which is directly related to the statistical power. Without controlling these confounding factors, the number of differentially expressed genes is not that informative.

2) Related to point #1 above, the authors did not filter differentially expressed genes based on fold change (i.e., effect size), except in Figure 2—figure supplement 1A (the cut off FDR<0.1 is also not very stringent). By comparing Figure 2 and Figure 2—figure supplement 1A, it seems that the vast majority of genes have quite moderate changes (|log2(fold change|<0.1). Given the large variation between replicates (e.g., for the GF-SI group as mentioned by the authors), the robustness of the results is questionable.

3) In particular, since changes in splicing factor expression is a central point of the paper, this reviewer is surprised that no specific splicing factors were discussed in any detail or even mentioned. At least some of these should be validated using independent samples by qRT-PCR and western blots.

4) The authors reported a large number of changes in exon usage. This reviewer noticed that DEXSeq was used for this analysis and I assume the software does not explicitly consider whether alternative splicing really occurs using exon junction reads. How many of the reported exons are really alternative exons (e.g., based on existing annotations)? Again, RT-PCR validation using independent samples will be necessary to support the claim.

5) Does the changes of any splicing factors explain the changes in exon usage? If yes, that will be helpful to argue for bona fide, causative changes.

6) The authors did not provide sufficient details on quality controls about the RNA-seq data, e.g., RIN number of the RNA samples, sequencing depth, mapping rate, the number of exonic reads, 3' bias, etc. These metrics are important to judge the quality of the data.

---

## [Author Response]

Essential revisions:

1) Molecular validation of key differentially expressed genes identified by transcriptomic analysis. Reviewers raised the concern that the manuscript is highly descriptive and lacks qPCR validation of key differentially expressed genes.

We have now added validation of key differentially expressed genes by qPCR, also several genes involved in MAP-K signalling (Figure 2—figure supplement 2). These data confirm that we observe increased baseline activity of several markers of neuronal activity. We also have to stress the point that we do see increased expression of MAP-K-related genes in the GF group albeit enrichment of these genes in response to stimulation is far less prominent as compared to the control group.

In addition, we have now discussed the fact that we do find exactly the expected regulations in the control group, which serves as a positive control and is internal validation of the behavioural stimulation protocol and the sequencing/bioinformatic analysis pipeline, i.e. the methodology of this paper. Also we want to highlight the fact we here find very similar results for the naïve comparisons as in our previous, independent report (Stilling et al., 2015).

2) Examination of within-group gene expression changes relative to behavioral performance (e.g., comparison between germ-free mice that demonstrated social interaction behaviours similar to controls and those that clearly showed a deficit).

A detailed examination of within-group gene expression in correlation to behavioural performance is provided in Figure 5 and Supplementary file 7 and respective Results and Discussion sections of the original manuscript, suggesting that indeed activation of the splicing machinery is the mechanism separating animals based on behavioral performance.

In addition, we have now provided additional within-group analysis of gene expression, comparing germ-free animals with high social behavior performance (GF-SI^high^) and animals with low social behavior performance (GF-SI^low^).

The description and data can be found in additional panels to Figure 5 (E)), a new sheet in Supplementary file 1 and the respective Results section. We used DESeq2 to perform within group comparison. Using a classical cut-off/threshold-based comparison, we find 3 genes significantly higher expressed in GF-SI^high^ animals. Using non-cut-off-based gene-set enrichment analysis (GSEA) we find several significantly enriched pathways and biological functions associated with RNA processing to show higher expression in GF-SI^high^ animals.

The new analysis thus confirms that genes of the splicing machinery are higher expressed in the amygdala of mice with control-like behavioural performance (GF-SI^high)^ in the social interaction test.

3) Examination of data relative to effect size of differential expression.

We have now provided additional analyses using gene-set enrichment analysis (GSEA) to avoid relying purely on p-value and/or fold-change cut-off thresholds. GSEA analyses ranked lists of genes, here sorted by fold-change as calculated by DESeq2 but includes all expressed genes for searching for enrichment among higher or lower expressed genes between two groups.

We have performed this analysis already before submitting our manuscript to *eLife*, but felt it would be confusing for readers as we find essentially the same biological functions and pathways as with our initial, classical cut-off-based analysis, which tends to be more stringent/conservative. We have therefore placed the new data in new Supplementary file 8 and comment only very briefly on the data in the revised manuscript.

4) Comment on disagreement between behavioral phenotype reported in this study as compared to previous report in Desbonnet et al., 2014. The reviewers raised the concern that the germ free mice do not show impairment in social behavior in this study, as compared to previous findings from the same group.

We have now commented on the differences between this study and Desbonnet et al., 2014 and extended the Discussion also on differences to other similar findings.

In short, our results are not in disagreement with our lab's previous report (Desbonnet et al., 2014). Contrarily, we confirm the finding by Desbonnet et al. that germ-free mice show reduced social interaction in the three-chamber social interaction task and that colonisation after weaning can reverse this effect. This is also in agreement with Buffington et al., 2016. In fact, only 2 out of 12 GF mice perform at average control levels. Natural variability in this outbred mouse strain (Swiss Webster), different experimenters (male vs. female, compare ref. Sorge, RE (2014) Nat Meth.) and a slightly different setup may together have contributed to greater variability in GF mice – a finding that actually helped us stratify the transcriptional response and correlate gene expression to behavioural phenotype (see Figure 5 and corresponding Results/Discussion sections)

5) Discussion of specific splicing factors in the context of existing literature and broader implications, and comment on whether changes in any splicing factors can explain the changes in exon usage.

Reanalysing the lists of genes with RNA processing functions, we find genes implicated in all steps of mRNA processing ranging from pre-mRNA generation to nuclear export and translational initiation. These include genes coding for proteins that bind RNA at cap and poly-A-tail (and respective enzymes catalysing capping and polyadenylation in the first place) as well as for protein components of all spliceosomal subunits and several representatives of dead-box binding protein (Ddx), Hnrnp, Rbm and Prfp families, all involved in nuclear or cytoplasmic RNA processing, guiding and surveillance. In addition as stated in the manuscript already, we find several non-coding RNAs involved in these processes such as the constituents of Caja bodies or paraspeckles, which are sites of active RNA processing. It is thus far beyond the scope of this study to provide a genome-wide matching of specific splicing factors and alternative exon-usage outcomes. In conclusion the data strongly suggests that these substantial changes in expression of the splicing machinery are causative for the observed differences in exon usage.

6) Additional methodological details of quality controls for sequencing, such as RIN number of the RNA samples, sequencing depth, mapping rate, the number of exonic reads, 3' bias, etc. are needed to evaluate the quality of the data. Also there is question as to whether the analytical software used considers whether alternative splicing occurs using exon junction reads. Consideration of how many of the reported exons are alternative exons based on existing annotations is needed.

We have now added an additional Supplementary file 8 with quality control details for RNA extraction and sequencing including a gene discovery curve as a quantitative measure of sequencing depth (number of reads vs. number of genes covered by these reads). On average we received 23.4 million (2.34x107) paired reads per sample, which covers most detectable genes. We have also uploaded FASTQC data as a ZIP file for review purposes, we leave it to the editor if this information needs to be published (this would be rather unusual).

Also see response to reviewer #3, pt. 4.

7) More nuanced discussion of the implications of findings. The reviewers request a more focused discussion of conclusions that can be drawn from the provided data.

While we do feel that this work needs to be put into a greater context, we have now provided a more focused discussion, restricted to the direct implications of the presented data. However, some of the other essential revisions required extending some parts of the Discussion (see above).

Reviewer #1:

Stilling et al. employed transcriptomics to determine the influence of the gut microbiome on social-interaction-induced changes in gene expression in the amygdala. By comparing naïve vs. socially stimulated mice, both control and germ-free (GF) mice, the authors found that social interaction triggers a differential change in gene expression in GF mice compared to control mice. Changes in the transcriptional landscape and GF social deficits were partially reversed by colonization at weaning. The authors identified significant overlap in a core population of genes that were differentially expressed in both naïve GF animals and socially stimulated controls. Among these, genes encoding elements of the MAPK pathway, which regulates neural plasticity, were upregulated in naïve GF animals, but only after social stimulation in conventionally colonized controls. Moreover, social stimulation markedly upregulated genes involved in RNA processing and splicing in GF animals, which was not observed in either control or GF animals colonised at weaning. Finally, the authors determined that the extent of alternative splicing observed in the amygdala of an individual correlated with the severity of the behavioral deficit observed in the animal.I find the paper by Stilling et al. to be an interesting piece of work. However, as it stands, the paper suffers from several deficiencies. First, the work is primarily descriptive and there is no molecular validation of the bioinformatic results.

While our study design does provide an intervention that contributes to causal inference (gene expression changes in the amygdala caused by social interaction), we agree that our work is primarily descriptive. We have now provided additional molecular validation of the bioinformatic results (Figure 2—figure supplement 2). In addition we want to point out the fact that we do find exactly the expected regulations in the control group (CON-SI), which is internal validation of the behavioural stimulation protocol, the methods and the sequencing pipeline. Also we do find very similar results for the naïve comparisons as in our previous, independent report (Stilling et al., 2015, BBI). We therefore agree with this reviewer that this issue is less problematic (see next point below).

Second, and perhaps more problematic, the GF mice do not show impairment in social behavior. Briefly, GF mice spent more time interacting with a mouse vs. an inanimate object, which is inconsistent with previous findings from the same group.

We have now commented on the differences between this study and Desbonnet et al., 2014 and extended the Discussion also on differences to other similar findings.

In short, our results are not in disagreement with our labs previous report (Desbonnet et al., 2014). Contrarily, we confirm the finding by Desbonnet et al. that germ-free mice show reduced social interaction in the three-chamber social interaction task and that colonisation after weaning can reverse this effect. This is also in agreement with Buffington et al., 2016. In fact, only 2 out of 12 GF mice perform at average control levels. Natural variability in this outbred mouse strain (Swiss Webster), different experimenters (male vs. female, compare ref. Sorge, RE (2014) Nat Meth.) and a slightly different setup may together have contributed to greater variability in GF mice – a finding that actually helped us stratify the transcriptional response and correlate gene expression to behavioural phenotype (see Figure 5 and corresponding Results/Discussion sections). Also see response to essential revisions above. We now include a discussion to clarify to what extend our data matches previous reports.

Finally, it is not clear why the study focuses exclusively on the amygdala. A more thorough study comparing gene expression pattern from brain regions involved in social behaviors will strengthen this study. My specific comments are below:

In addition to the justification already mentioned in the paper, we have now rephrased the respective section in the Introduction and added specific references to justify our choice. We certainly do not imply that this is the only relevant region in this context, as we have previously studied also, e.g., the prefrontal cortex and have made similar observations with regard to a potential neuronal hyperactivity (Hoban et al. 2016, Transl. Psych.).

1) The data presented in Figures 1B and C are inconsistent with a previous study from this group (Desbonnet et al., 2014) in which GF animals preferred interaction with an empty cup over interaction with a mouse in the 3-chamber task. This inconsistency should be addressed and the data in panels B and C should be combined into a single panel with accompanying statistical data from both intra- and inter-group two-way ANOVAs.

See response to general comment above. Figure 1B-C has been revised to reflect this recommendation.

2) The authors should validate the top 10-25 transcripts reported in Figure 2 by qPCR or Western blotting and provide these data in the figure. In the same line, MAPK pathway activity is readily accessible for analysis by Western blotting. Can the authors provide molecular validation of the MAPK pathway data-baseline upregulation in GF animals and post-interaction upregulation in controls?

We have now added validation results for key differentially expressed genes by qPCR, also several genes involved in MAP-K signalling. These data confirm that we observe increased baseline activity of several markers of neuronal activity. We also have to stress the point that we do see increased expression of MAP-K-related genes in the GF group albeit enrichment of these genes in response to stimulation is far less prominent as compared to the control group.

In addition, we have now added a few words to discuss the fact that we do find exactly the expected regulations in the control group, which serves as a positive control and is internal validation of the behavioural stimulation protocol and the sequencing/bioinformatic analysis pipeline, i.e. the methodology of this paper. Also we want to highlight the fact we here find very similar results for the naïve comparisons as in our previous, independent report (Stilling et al., 2015).

3) It's unlikely that the alternative splicing events in GF animals are restricted to the amygdala. Other structures including those in the mesolimbic dopaminergic reward system, specifically the nucleus accumbens and ventral tegmental area, are emerging as regions critically involved in regulating social behavior. Did the authors perform similar transcriptomic analysis on tissue isolated from other brain regions following the interaction? This data would bolster the authors' claim that colonization-state dependent alterations in RNA splicing and processing exerts meaningful influence over social behavior.

We agree with this reviewer that more data from other brain regions would further advance the field. In fact, we do provide gene expression data in GF mice also in the prefrontal cortex in another published study (Hoban et al. 2016, Transl. Psych). Unfortunately, in this study we had to focus on a particular brain region that is a critical node in microbiome to brain signalling (see Cowan et al. Bioessays 2018 http://onlinelibrary.wiley.com/doi/10.1002/bies.201700172/abstract) and thus we feel that repeating this study in multiple other brain regions is beyond the scope of this paper.

4) The lengthy Discussion presents an interesting theory unifying neuroimmune interaction data with findings from the microbiome and behavior field; however, this is completely outside of the scope of the conclusions that can be drawn from the provided data and, in my view, the implications of this study should be toned down.

While we do feel that this work needs to be put into a greater context, and provides a coherent mechanistic understanding of the processes described that has – to our knowledge – not been presented before, we have now focussed the Discussion, restricted to the direct implications of the presented data. However, some of the other essential revisions required extending some parts of the Discussion (see above).

Reviewer #2:

[…] 1) Not sure if comparisons with behaviour naïve and tested mice really add to the study, and perhaps can be moved to supplementary data? The work is not addressed in the Discussion anyway – what key information can the reader glean from these associations?What really would have been interesting to see are comparisons between the GF mice that demonstrated social interaction behaviours similar to controls (>300 sec), with those that clearly showed a deficit (<200sec) in Figure 1B i.e. what are the molecular profiles within the GF-SI cohorts? The authors did mention variability amongst these animals, so are these variabilities reflected at the molecular level? Obviously a meaningful analysis might be under-powered, but even a cursory inspection would be valuable.

A detailed examination of within-group gene expression in correlation to behavioural performance is provided in Figure 5 and Supplementary file 7 and respective results and Discussion sections of the original manuscript, suggesting that indeed activation of the splicing machinery is the mechanism separating animals based on behavioural performance.

In addition, we have now provided further within-group analysis of gene expression, comparing germ-free animals with high social behaviour performance (GF-SI^high^) and animals with low social behaviour performance (GF-SI^low^).

The description and data can be found in additional panels to Figure 5 (E), a new sheet in Supplementary file 1 and the respective Results section. We used DESeq2 to perform within group comparison. Using a classical cut-off/threshold-based comparison, we find 3 genes significantly higher expressed in GF-SI^high^ animals. Using non-cut-off-based gene-set enrichment analysis (GSEA) we find several significantly enriched pathways and biological functions associated with RNA processing to show higher expression in GF-SI^high^ animals.

The new analysis thus confirms that genes of the splicing machinery are higher expressed in the amygdala of mice with control-like behavioural performance (GF-SI^high^) in the social interaction test.

2) Subsection “Social interaction induces highly distinct gene expression patterns in the amygdala of germ-free mice”, first paragraph: results should be presented immediately, perhaps with an introductory sentence – the preceding detailed paragraph with references is too excessive – this should be moved to Materials and methods or Introduction – but ideally, as mentioned above, these comparisons should be hidden in the shadows so the reader can focus on the main findings between CON-SI, GF-SI & exGF-SI!

We fully agree with this reviewer’s excitement for the novel data and have radically shortened this part. Only a rough description of the sequencing conditions has remained as we believe this is important information for readers.

Reviewer #3:

Stilling et al. hypothesized that social behavior can be modulated by host-associated microbiota. To test this hypothesis, the authors compared control mice, germ-free mice and GF mice colonized post weaning by RNA-seq analysis of amygdala upon exposure to social interaction. The authors found significant gene expression changes between groups. In particular, GF mice with SI exposure showed increased expression of splicing factors, as well as differential splicing of a large number of exons. While the hypothesis is potentially interesting, this reviewer is concerned whether the analysis has properly controlled and reproducible to make the claims. Overall, the work is also quite descriptive in general.1) The authors judge the difference of two conditions by the number of differentially expressed genes. This is problematic because the number of differentially expressed genes is largely affected by the number of replicates and sequence depth, which is directly related to the statistical power. Without controlling these confounding factors, the number of differentially expressed genes is not that informative.

We have now provided gene-set enrichment analysis (GSEA), which is not affected by potentially biased cut-offs for p-value and fold-change.

2) Related to point #1 above, the authors did not filter differentially expressed genes based on fold change (i.e., effect size), except in Figure 2—figure supplement 1A (the cut off FDR<0.1 is also not very stringent). By comparing Figure 2 and Figure 2—figure supplement 1A, it seems that the vast majority of genes have quite moderate changes (|log2(fold change|<0.1). Given the large variation between replicates (e.g., for the GF-SI group as mentioned by the authors), the robustness of the results is questionable.

We have now provided gene-set enrichment analysis (GSEA), which is not affected by potentially biased cut-offs for p-value and fold-change. In addition we want to point out the fact that we do find exactly the expected regulations in the control group, which is internal validation of the behavioural stimulation protocol, the methods and the sequencing pipeline. Also we do find very similar results for the naïve comparisons as in our previous, independent report (Stilling et al., 2015).

3) In particular, since changes in splicing factor expression is a central point of the paper, this reviewer is surprised that no specific splicing factors were discussed in any detail or even mentioned. At least some of these should be validated using independent samples by qRT-PCR and western blots.

See response to essential revision #5.

4) The authors reported a large number of changes in exon usage. This reviewer noticed that DEXSeq was used for this analysis and I assume the software does not explicitly consider whether alternative splicing really occurs using exon junction reads. How many of the reported exons are really alternative exons (e.g., based on existing annotations)? Again, RT-PCR validation using independent samples will be necessary to support the claim.

We have extensive experience with using the described pipeline and are confident that our methods adhere to the highest standards. In fact, the number of differentially spliced genes was exceptionally high in the GF-SI and exGF-SI comparisons based on many previous datasets generated and analysed by the authors of this paper.

Regarding using exon junction reads, the DEXSeq publication states:

“Junction reads are reads whose genomic alignment contains a gap because they start in one exon, end in another exon, and “jump” over the intron in between and possibly over skipped exons. In DEXSeq, such reads are counted for each counting bin with which they overlap, i.e., they appear multiple times in the count table. However, because we test for each exon separately, this does not affect the validity of the test. […] Furthermore, junction reads give evidence for connections between counting bins and thus are crucial for isoform deconvolution tools such as Cufflinks and MMSeq. For our exon-by-exon test, however, leveraging this information is not essential, and also not straightforward. In the method presented, we essentially consider for each sample the ratio of the number of reads overlapping with an exon to the number of reads falling onto the whole gene.”

In addition for most genes the dominant isoform is not determined and may be different in different cell types or tissues. Therefore the definition of what is an “alternative exon” is not straight forward and we feel that it can thus not feasibly be included in this paper.

5) Does the changes of any splicing factors explain the changes in exon usage? If yes, that will be helpful to argue for bona fide, causative changes.

Reanalysing the lists of genes with RNA processing functions, we find genes implicated in all steps of mRNA processing ranging from pre-mRNA generation to nuclear export and translational initiation. These include genes coding for proteins that bind RNA at cap and poly-A-tail (and respective enzymes catalysing capping and polyadenylation in the first place) as well as for protein components of all spliceosomal subunits and several representatives of dead-box binding protein (Ddx), Hnrnp, Rbm and Prfp families, all involved in nuclear or cytoplasmic RNA processing, guiding and surveillance. In addition as stated in the manuscript already, we find several non-coding RNAs involved in these processes such as the constituents of Cajal bodies or paraspeckles, which are sites of active RNA processing. It is thus far beyond the scope of this study to provide a genome-wide matching of specific splicing factors and alternative exon-usage outcomes. In conclusion the data strongly suggests that these substantial changes in expression of the splicing machinery are causative for the observed differences in exon usage.

6) The authors did not provide sufficient details on quality controls about the RNA-seq data, e.g., RIN number of the RNA samples, sequencing depth, mapping rate, the number of exonic reads, 3' bias, etc. These metrics are important to judge the quality of the data.

We have now added an additional Supplementary file 8 with quality control details for RNA extraction and sequencing including a gene discovery curve as a quantitative measure of sequencing depth (number of reads vs. number of genes covered by these reads). On average we received 23.4 million (2.34x10^7^) paired reads per sample, which covers most detectable genes.